# Nutraceutical Potential of *Havardia pallens* and *Vachellia rigidula* in the Diet Formulation for Male Goat

**DOI:** 10.3390/metabo15070457

**Published:** 2025-07-05

**Authors:** Jesús Humberto Reyna-Fuentes, Cecilia Carmela Zapata-Campos, Jorge Ariel Torres-Castillo, Daniel López-Aguirre, Juan Antonio Núñez-Colima, Luis Eliezer Cruz-Bacab, Fabián Eliseo Olazarán-Santibáñez, Fernando Sánchez-Dávila, Aida Isabel Leal-Robles, Juan Antonio Granados-Montelongo

**Affiliations:** 1Facultad de Medicina Veterinaria y Zootecnia, Universidad Autónoma de Tamaulipas, Carretera Victoria-Mante Km 5, ejido Santa Librada, Ciudad Victoria 87274, Tamaulipas, Mexico; jesushumbertoreyna@gmail.com (J.H.R.-F.); feolazaran@docentes.uat.edu.mx (F.E.O.-S.); 2Instituto de Ecología Aplicada, Universidad Autónoma de Tamaulipas, Colonia División del Golfo 356, amp la Libertad, Ciudad Victoria 87019, Tamaulipas, Mexico; joatorres@docentes.uat.edu.mx; 3Facultad de Ingeniería y Ciencias, Universidad Autónoma de Tamaulipas, Campus del Centro Universitario, Ciudad Victoria 87149, Tamaulipas, Mexico; dlaguirre@docentes.uat.edu.mx; 4Departamento de Recursos Naturales Renovables, Universidad Autónoma Agraria Antonio Narro, Calzada Antonio Narro 1923, Buenavista, Saltillo 25315, Coahuila, Mexico; juan.anunezc@uaaan.edu.mx (J.A.N.-C.); juanantonio.granados@gmail.com (J.A.G.-M.); 5División Académica de Ciencias Agropecuarias, Universidad Juárez Autónoma de Tabasco, Carretera Villahermosa-Teapa, Km 25, R/A. La Huasteca 2a Sección, Villahermosa 86280, Tabasco, Mexico; lecb82@gmail.com; 6Facultad de Agronomía, Universidad Autónoma de Nuevo León, Posgrado Conjunto, Francisco I. Madero s/n, Hacienda la Cañada, General Escobedo 66050, Nuevo Leon, Mexico; fernando_sd3@hotmail.com; 7Departamento de Botánica, Universidad Autónoma Agraria Antonio Narro, Calzada Antonio Narro 1923, Buenavista, Saltillo 25315, Coahuila, Mexico; aisaler@yahoo.com.mx

**Keywords:** secondary metabolites, polyphenols, antioxidants, diet, scrub agroecosystems

## Abstract

**Background**: Xerophilous scrubland is a semi-desert ecosystem characterized by a wide diversity of shrubs, which have secondary compounds with nutraceutical potential that could be used as feed for livestock, specifically by goats, since this species has developed behavioral and physiological adaptations that allow it to take advantage of the plant resources of said scrubland. **Objective**: To evaluate the nutraceutical potential of *Havardia pallens* and *Vachellia rigidula*, native species of the xerophilous scrubland, when incorporated as ingredients in goat diets. **Methods:** Integral diets for male goats were prepared, formulated with 35% inclusion of *Havardia pallens*, *Vachellia rigidula*, and *Medicago sativa*, the latter used as a plant control species. The content of flavonoids and total phenols was compared using colorimetric methods, and the antioxidant capacity was measured using the FRAP method. RP-HPLC-ESI-MS characterized the bioactive compounds in the different extracts. Statistical analysis was performed by ANOVA. **Results**: The aqueous extraction of *Vachellia rigidula* showed the highest concentration of total phenols ($\bar{x}$ = 18.22 mg GAE/g^−1^), followed by the ethanolic extract in the same species ($\bar{x}$ = 17.045 mg GAE/g^−1^). Similarly, *Vachellia rigidula* presented the highest antioxidant capacity ($\bar{x}$ = 144,711.53 µmol TE/g^−1^), while *Medicago sativa* presented the lowest ($\bar{x}$ = 11,701.92 µmol TE/g). The RP-HPLC-ESI-MS analysis revealed that *Vachellia rigidula* presented a higher abundance of flavones, catechins, flavonols, methoxyflavones, and tyrosols. However, *Harvardia pallens* presented higher levels of methoxycinnamic and hydroxycinnamic acids. One-way ANOVA results showed that diets containing 35% *Vachellia rigidula* and *Havardia pallens* significantly contrasted (*p* < 0.05), increased the content of secondary compounds and antioxidant capacity compared to the control species. Furthermore, including *Vachellia rigidula* led to a significantly higher antioxidant capacity (*p* < 0.05) than diets with *Havardia pallens* or *Medicago sativa*. **Conclusions**: Incorporating the leguminous shrubs *Vachellia rigidula* and *Havardia pallens* into the formulation of comprehensive diets for buck goats improves the content and availability of phenols, flavonoids, and antioxidants. However, in vivo evaluation of these diets is important to determine their physiological and productive effects on the animals.

## 1. Introduction

The Food and Agriculture Organization of the United Nations (FAO) suggests the development and implementation of effective strategies for food security, one of which is to provide society with livestock species that can produce efficiently under adverse environmental conditions, thus obtaining an adequate and sustainable use of local resources [1]. In this context, goat farming has great development potential, since this animal species has proven capable of adapting to adverse environmental conditions such as climate, availability and distribution of plant resources, plant characteristics, and water availability, characteristics that other livestock species, such as cattle, cannot develop [2,3]. Goat feeding is based on obtaining the necessary elements from the rangeland for their growth, development, and reproduction, such as proteins, fiber, vitamins, minerals, and energy [4]. However, when these elements are not adequately present in the diet, goat farmers usually supplement them with grain concentrates and the purchase of forage legumes. For small producers, supplementing the diet with grains or forage hay is expensive and depends largely on its availability in the region and throughout the year [5].

The xerophile scrubland in Mexico, specifically the Tamaulipas thornscrub, is characterized by a wide diversity of plant species, mainly shrubs that can be used as fodder for goats, since they have nutritional and chemical characteristics that could have a positive effect on meat and milk production. In this sense, the use of forage shrubs could be implemented as a viable strategy to be added to comprehensive diets and thus improve feed quality and reduce production costs by partially or fully replacing cereal grain-based feeds using local plant resources [6]. *Vachellia rigidula* and *Havardia pallens* are two shrub species that could be used as fodder for goats. Both are native to the Tamaulipas thornscrub ecosystem. *Vachellia rigidula* is a shrub or small tree with a crude protein (CP) content of 15%, neutral detergent fiber (NDF) of 49.3%, acid detergent fiber (ADF) of 36.5%, and cellulose of 26.4% [7]. On the other hand, *Harvardia pallens* is characterized by higher leaf renewal between April and May; however, its mature leaves remain throughout the year, even during winter. This species favors the development of herbaceous species such as *Lantana* spp. and *Stipa lessingiana* (grass), which also have forage potential [8]. Both shrub species are consumed by goats grazing in rangelands of northeastern Mexico [9]. Shrubs often contain bioactive compounds (e.g., flavonoids, terpenes, phenols, etc.) produced by the plant from secondary metabolism [10]. Based on these observations, the present research sought to evaluate the nutraceutical potential of *Havardia pallens* and *Vachellia rigidula* when incorporated as ingredients in diets formulated for male goats.

## 2. Materials and Methods

### 2.1. Study Site

This study was developed in a semiarid zone of Tamaulipas; specifically, in the hydrological basin of the Purificación River and between the foothills of the Sierra Madre Oriental. Its location can be found between 23°44′06″ N and 99°07′51″ W at an altitude of 321 m and intersected by the Tropic of Cancer at 23°27′. The zone has a mean annual temperature of 23.5 °C and the mean annual precipitation is 780 mm, with a semi-dry and warm sub-humid climate [11].

### 2.2. Plant Collection

Plant samples were collected from the Boca de Juan Capitán common land in Victoria, Tamaulipas, Mexico. This area was selected for the abundance of *V. rigidula* (gavia) and *H. pallens* (tenaza) at a height below 2 m. Shrubs were randomly selected, with three leaf and stem samples taken from each species. The forage legume *M. sativa* (alfalfa) was selected as a control plant. It should be noted that this legume is not native to the region and was therefore purchased from a forage company. Samples were stored at −18 °C until processing.

### 2.3. Bromatological Analysis

The samples were analyzed in the animal nutrition laboratory of the Faculty of Engineering and Sciences of the Autonomous University of Tamaulipas. The plant species samples were dehydrated in a forced-air oven at 50–60 °C for 48 h to determine the partial dry matter (DM). The dried samples were ground in a Wiley mill with a 1 mm sieve (Model 4; Arthur H. Thomas Co., Philadelphia, PA, USA). Bromatological analysis was performed in triplicate. The ash content was determined (ID 942.05) by incineration at 600 °C for 2 h using a muffle furnace. Crude protein (CP) (ID 954.05) was obtained using the macro-Kjeldahl method (N × 6.25) [12]. The fibrous fraction was determined through acid detergent fiber (ADF), neutral detergent fiber (NDF), and lignin using the procedure described by Van Soest [13].

### 2.4. Metabolite Extraction and Secondary Compound Analysis

An aqueous extraction of three sub-samples (5 g each) from the ground extract of the three plant species was carried out. The samples were weighed in 100 mL Erlenmeyer flasks, then poured with 50 mL of distilled water at 60 °C and homogenized. The samples were then placed in an oven at 60 °C, stirring every 15 min for 60 min. The samples were filtered through a Whatman membrane (No. 41). To obtain a larger amount of extract, the samples were centrifuged for 20 min at 3500 rpm. Finally, the aqueous extract was placed in amber bottles and stored at a temperature of 4 °C until processing. The methanolic and ethanolic extraction was carried out by weighing 2.5 g of each sample and placing it in a test tube to which 25 mL of each of the solutions and distilled water (70:30 *v*/*v*) were added, it was vortexed to homogenize and left to stand for 24 h, avoiding exposure to light and refrigerated at 4 °C. Subsequently, it was centrifuged for 20 min at 3500 rpm and the supernatant was obtained for later analysis.

### 2.5. Total Flavonoids

The content of total flavonoids was determined as previously described [14]; 10 µL aliquot of the supernatant of the triplicate extracts, prepared previously, was used, 1.5 mL of ethanol (IBI Scientific, catalog no. 15720, Dubuque, IA, USA.) at 95%, 2.7 µL of an AlCl_3_ solution (R.A article 24011-F, key H87), 2.7 µL of 1 M CH_3_CO_2_K solution (catalog 5089, CAS No. 127-08-2) and 102.7 µL of distilled water (REPR brand, article 10294, key H87) will be added; the mixture will be incubated for 40 min. Readings were taken using a spectrophotometer (model no. iMark, serial no. 11293) at 415 nm wavelength. The quantification of the concentration was performed using a standard curve prepared with quercetin (SIGMA brand, item Q4951-10G). Results were expressed as mg quercetin equivalents per g dry matter (mg/EQ/g^–1^ DM).

### 2.6. Total Phenols

Total phenols were determined by the Folin Ciocalteu technique [15]; 2 µL of the extracts were used in triplicate, deposited in a 96-microwell plate, then 25 µL of the Folin Ciocalteu reagent (Sigma Aldrich F9252) was added, homogenized and let stand for 5 min, then 125 µL of anhydrous sodium carbonate (catalog 5015, CAS No. 497-19-8 (NaCO_3_, 0.01 M) was added, stirred again and let stand for 5 min. Finally, 49 µL of distilled water was added to perform the incubations and to be read in a spectrophotometer (model No. iMark, series 11293) at 750 nm. The concentration of total phenols was calculated using gallic acid, and the results were expressed as mg of gallic acid equivalent per gram of DM of the plant extract (mg/EAG/g^−1^ DM).

### 2.7. Antioxidant Capacity by the Ferric Iron (Fe^+3^) Reduction Method (FRAP)

The FRAP method is based on the reduction of ferric iron (Fe^+3^) present in the FRAP reagent to the ferrous form (Fe^+2^) by the presence of antioxidants [16]. For the reading of the samples, 900 µL of FRAP solution, 30 µL of sample, and 120 µL of distilled water were used. The FRAP solution is composed of 25 mL of acetic acid-sodium acetate buffer solution (pH 3.6), 2.5 mL of 10 [mM] TPTZ solution diluted with 40 [mM] HCl and 20 [mM] FeCl3 solution. A blue coloration is generated, of intense proportionality to the reducing capacity of the sample (a ferrous-TPTZ complex is generated) that can be quantified by colorimetry (593 nm) based on a ferrous sulfate standard. The absorbance was determined at a wavelength of 593 nm in a spectrophotometer reader model No. iMark, series 11293. For each reading, the absorbance reading of the control sample was considered. The final absorbance of the samples was compared with the standard curve of Trolox (100–1000 μmol/L) dissolved with ethanol (brand IBI SCIENTIFIC, catalog IBI 15720 at 96%).

### 2.8. RP-HPLC-ESI-MS

The reverse phase-high performance liquid chromatography analyses were performed in a Varian HPLC system including an autosampler (Varian ProStar 410, USA), a ternary pump (Varian ProStar 230I, USA), and a PDA detector (Varian ProStar 330, USA). A liquid chromatograph ion trap mass spectrometer (Varian 500-MS IT Mass Spectrometer, USA) equipped with an electrospray ion source was also used. In brief, 10 µL of the samples were injected into a Denali C18 column (150 mm × 2.1 mm, 3 µm, Grace, USA). The oven temperature was maintained at 30 °C. The eluents were formic acid (0.2%, *v*/*v*; solvent A) and acetonitrile (solvent B). The following gradient was applied: initial, 3% B; 0–5 min, 9% B linear; 5–15 min, 16% B linear; 15–45 min, 50% B linear. The column was then washed and reconditioned. The flow rate was maintained at 0.2 mL/min, and elution was monitored at 245, 280, 320, and 550 nm. The whole effluent (0.2 mL/min) was injected into the source of the mass spectrometer without splitting. All MS experiments were conducted in negative mode [M-H]^−1^. Nitrogen was used as nebulizing gas, and helium as damping gas. The ion source parameters were spray voltage 5.0 kV, capillary voltage 90.0 V, and temperature 350 °C. The data were collected and processed using MS Workstation software (V 6.9). The samples were first analyzed in full scan mode, acquired in the *m*/*z* range 50–2000 [17,18].

### 2.9. Diet Formulation and Standardization

The protein and energy levels of the diets were standardized based on the nutritional requirements of growing male goats [19]. Inclusion calculations were made based on the nutritional characteristics of each raw material, ground sorghum (*Sorghum bicolor*), sorghum straw (dry vegetative residues from the sorghum crop, typically composed of stems and leaves, commonly used as a fibrous source in ruminant diets), molasses, soybean meal (*Glycine max*), *M. sativa, H. pallens*, and *V. rigidula*) concerning DM, CP, EE, NDF, and ADF. The diets were then balanced (DME, a diet with *M. sativa*; DHP, a diet with *H. pallens*; DVR, a diet with *V. rigidula*) using a trial-and-error method. To prepare the diet, the ingredients were weighed and mixed using a homogenizer (Oster FPSTSM3711). For shrubs, leaves were used, while fibrous materials such as sorghum and alfalfa straw were ground to facilitate mixing. Water was added to the liquid ingredient, molasses (70:30 *v*/*v*), and incorporated during homogenization. The same procedure was followed, but without the inclusion of the plants, to a mixture of only the conventional ingredients, thus obtaining a fourth diet. The samples were then dehydrated in a forced-air oven at 50–60 °C for 48 h to determine partial dry matter (DM). Subsequently, they were ground in a Wiley mill equipped with a 1 mm sieve (Model 4; Arthur H. Thomas Co., Philadelphia, PA, USA). The diets were stored until further analysis.

### 2.10. Statistical Analysis

All analyses were conducted with the analytical software package SAS version 9.2 [20]. A one-way analysis of variance (ANOVA) was performed for each evaluated variable, including bromatological parameters, total phenol and flavonoid concentrations, and antioxidant capacity. When significant differences were identified (*p* < 0.05), means were compared using Tukey’s post hoc test.

## 3. Results

### 3.1. Bromatological Analysis

Table 1 shows the diets formulated with 35% inclusion of the native shrubs *H. pallens* and *V. rigidula*, as well as the forage legume *M. sativa*.

The bromatological analysis (Table 2) of the plant species determined that *V. rigidula* had the highest FDA value (40.48 ± 4.71%), followed by *H. pallens* with 31.43 ± 4.75%. Regarding NDF, no significant differences were observed between the plant species evaluated. The level of fibrous fraction decreased in whole grain diets; however, the DVR diet made with *V. rigidula* leaves had the highest fibrous content of both FDA (21.18 ± 0.54%) and NDF (37.47 ± 0.90%). The plant species with the lowest fiber content was *M. sativa*; however, in whole grain diets, this element was similar to *H. pallens*. Regarding CP, *M. sativa* recorded the highest value (22.66 ± 2.13%), which was statistically comparable to that of *H. pallens* (20.67 ± 1.69%). The CP content decreased in integral diets; the DME diet had the highest protein content (16.7 ± 0.01%), while the DHP (14.24 ± 0.35%) and DVR (13.4 ± 0.05%) diets showed statistically similar CP levels. The ash content was significantly higher in the formulated diets than in the forage species. In this sense, the DME diet presented the highest ash content (11.86 ± 0.09%), followed by the DHP (11.33 ± 0.10%) and the DVR (10.15 ± 0.02%). Finally, regarding the lignin content, *V. rigidula* presented the highest value (21.4 ± 3.43%), significantly higher than that of the other plant species and the integral diets.

### 3.2. Total Phenolic (TPC) and Flavonoid Content (TFC)

The concentration of phenolic compounds differed significantly between plant species and extraction solvents (*p* < 0.01). However, the flavonoid content did not show significant variation (*p* > 0.05). The aqueous extract of *V. rigidula* exhibited the highest concentration of phenolic compounds ($\bar{x}$ = 18.22 mg/GAE/g^−1^), followed by its ethanolic extract ($\bar{x}$ = 17.05 mg/GAE/g^−1^). The lowest phenolic concentrations were recorded in the methanolic and ethanolic extracts of *M. sativa* ($\bar{x}$ = 7.61 mg/GAE/g^−1^) (Table 3).

The total concentration of secondary compounds in the experimental diets revealed that the DVR-based formulation had the highest total phenolic content per gram (g/GAE/g^−1^). However, this value was statistically similar to the DHP diet, while the DME diet had the lowest concentration (*p* = 0.007). Similarly, the total flavonoid content of the DME and DHP diets did not show significant differences. However, the DVR diet showed a significantly higher flavonoid concentration ($\bar{x}$ = 230/g/QE/g^−1^) (*p* < 0.05) (Table 4).

### 3.3. Antioxidative Capacity

The native species *V. rigidula* presented the highest level of antioxidant capacity (144,711.53 μmol /TE/g^−1^), while *H. pallens* was statistically similar to *M. sativa*. (Table 5).

The antioxidant capacity of diets formulated with *V. rigidula*, *H. pallens*, and *M. sativa* varied significantly depending on the type of extraction. Diets with *V. rigidula* (DVR) showed the highest antioxidant activity in both aqueous (147,966.66 ± 10,315.38 μmol/TE/g^−1^) and methanolic (139,847.44 ± 11,097.05 μmol/TE/g^−1^) extracts, being significantly higher (*p* < 0.05) than those formulated with *H. pallens* (DHP) and *M. sativa* (DME), whose antioxidant capacities were considerably lower (Table 6). When native species such as forage are eliminated from the diet, the antioxidant capacity is further reduced in the μmol/TE/g^−1^ values in the different extraction solvents (Table 6).

### 3.4. Identification of Phytochemicals

Twelve organic compounds were identified in the extracts of *Vachellia rigidula* through RP-HPLC-ESI-MS, including flavones, flavanones, methoxyflavones, catechins, proanthocyanidins, tyrosols, lignans, and hydroxycinnamic acids. Moreover, the phytochemical analyses of aqueous, ethanolic, and methanolic extracts of *V. rigidula* highlight the presence of compounds with great antioxidative potential (Table 7).

The phytochemical analysis of *Havardia pallens* revealed six secondary compounds classified within the groups of flavanones, flavonols, methoxycinnamic acids, hydroxycinnamic acids, methoxyflavones, and methoxyflavonols. In this manner, compounds such as Ferulic acid 4-O-glucoside, Caffeic acid 4-O-glucoside, Sinensetin, Myricetin, 5,3′,4′-Trihydroxy-3-methoxy-6:7-methylenedioxyflavone 4′-O-glucuronide, feruloyl glucose, Myricetin 3-O-rhamnoside and Peonidin 3-O-(6″-acetyl-glucoside) were also identified (Table 8).

## 4. Discussion

### 4.1. Bromatological Analysis

The nutritional status of animals is an important factor in determining the quality of their products and derivatives (e.g., meat, milk, dairy products). It can be improved by including secondary compounds such as phenolics, terpenes, and flavonoids in their diet [44,45]. In this context, animal feed can be strategically enriched with these compounds to provide a functional and nutraceutical supplement [46]

Functional foods, by definition, possess intrinsic nutraceutical value that, regardless of whether their active components are fully identified, contributes to maintaining and improving animal health. Furthermore, bioactive compounds can be administered in concentrated forms or incorporated directly into feed formulations, thus improving their functional properties [45].

Native shrub legumes, such as *Havardia pallens* and *Vachellia rigidula*, are available year-round in the Tamaulipas thornscrub ecosystem and are frequently consumed by browsing goats. However, *Medicago sativa*, particularly its regrowth paste (straw), is widely used as a ruminant feed resource [[9],[10],[11],[12],[13],[14],[15],[16],[17],[18],[19],[20],[21],[22],[23],[24],[25],[26],[27],[28],[29],[30],[31],[32],[33],[34],[35],[36],[37],[38],,[39],[40],[41],[42],[43],[44],[45],[46],[47]]. As nitrogen-fixing plants, legumes are typically characterized by their high protein content, which varies between 12% and 30%. The available information on the bromatological characteristics of *V. rigidula* is limited. However, Ramírez et al. (1999) [48] reported crude protein (CP), acid detergent fiber (ADF) and lignin values of 16.4%, 41.8% and 22.9% respectively; likewise Ramirez-Lozano (2006) [49] reported 17% of CP, 14% of lignin and 15% of cell wall, values that closely coincide for lignin and ADF, however for CP our value is 4.16 percentage points below. For *Havardia pallens*, Foroughbakhch et al. (2013) [9] reported a CP content of 24%. NDF, ADF, and lignin levels are critical indicators of feed potential and cell wall digestibility in ruminants. Ramírez-Orduña et al. (2002) [50] reported a lignin content of 7.4% for *Havardia pallens* and 17.4% for *Vachellia rigidula*. There is insufficient bromatological information on both species; however, in agronomic data, *Havardia pallens* is the species with the second highest relative abundance index in areas of regenerating scrub disturbed by livestock activity [51], and *Vachellia rigidula* also showed a strong correlation in the estimation of leaf biomass [52]. Therefore, these two species are abundant in the thorny scrubland of Tamaulipas, so they can provide food for goats at strategic times. Torres-Fajardo et al. (2021) [53] mention that to decide whether a plant is used as a functional food or nutraceutical, it is necessary to know if this plant is present in the rangeland and if the animals consume it. Both *Vachellia rigidula* and *Havardia pallens* meet these conditions.

Grazing goats naturally select plants that meet their nutritional needs at various physiological stages [54]. In this sense, the dry season diet of pregnant goats consists of 64% NDF, 43% ADF, and 7% lignin [55], while the diet of adult goats during the rainy season contains 54% NDF, 36% ADF, and 7% lignin [56]. Both studies indicate the values found are within acceptable ranges for fibrous fractions in ruminant diets.

Gholve et al. (2021) [57] formulated a diet for growing males and females, with an inclusion of 25% of the legume Sesbania sesban, yielding 19% CP, 49.2% NDF, and 41.8% ADF. This formulation allowed daily weight gains of 45.76 g/day, comparable to those obtained with commercial food (45.62 g/day). Zapata-Campos et al. (2021) [10] used leaves of *Leucaena leucochepala*, *Vachellia farnesiana*, and *Prosopis laevigata* as alternative ingredients in grain diets for growing male goats with a 33% inclusion, as well as a control diet based on alfalfa. They found that the average daily gain (GDP) did not differ between males fed alfalfa, *A. farnesiana*, or *L. leucocephala* (120 26, 134 37, and 103 29 g/d, respectively). Males fed alfalfa and *A. farnesiana* had larger scrotal circumferences (*p* < 0.05) (26.6 0.4 and 25.8 1.5 cm) than those fed *L. leucocephala* or *P. laevigata* (24.3 1.2 and 24.1 2.0 cm). They concluded that a conventional legume can be replaced by native legumes. Therefore, the legumes studied here, with the results obtained in the bromatological analysis, may be a suitable alternative ingredient for use in grain-based diets, especially to complement the diets of male goats.

### 4.2. Phenolic Acids

Goats have developed different adaptive mechanisms to tolerate the secondary compounds they consume from rangeland plant species. These are anatomical, such as the enlargement of the parotid glands to increase salivation, or physiological, such as the production of proline-rich salivary proteins (PRPs) with the ability to bind tannins [58,59]. Native legumes, in addition to their protein content, also contain secondary compounds [60] that can exert beneficial or detrimental effects on the animals that consume them [61].

The presence of phenolic compounds is often indicative of tannin content [61,62]. When consumed at high concentrations, tannins can interfere with biological functions by reducing crude protein (CP) utilization, decreasing feed palatability, and inhibiting the activity of various digestive enzymes [63,64]. However, when present at adequate levels, tannins can provide multiple functional benefits, including antioxidant, antibiotic, anti-inflammatory, and anthelmintic effects [65,66]. These effects can be observed in both animal performance and the quality of animal products [67]. The phenolic content in *Vachellia rigidula* and *Havardia pallens* is lower than that reported for other plant species, such as *Vachellia nubica* ($\bar{x}$ = 30 mg/GAE/g^−1^) and *Vitex negundo* ($\bar{x}$ = 27.72 mg/GAE/100 g^−1^), although it is higher than in *Moringa oleifera* ($\bar{x}$ = 7.18 mg/GAE/g^−1^) [68,69]. For his part, Arenaz (2021) [70] found 124.98 mg GAE/g in acetone extract, while in ethanolic extract, the phenol content was 125.93 mg GAE/g for *Vachellia rigidula* leaves, values higher than those found in our experiment. In general, the leaves of the *Acacia* or *Vachellia* genus have a significant presence of secondary compounds that have been used mainly in traditional medicine [71]. However, his effect is also reflected in animals since the plant species of this genus are very abundant in pastures grazed by goats [72].

Flavonoids, known for their antioxidant and free radical scavenging properties [73], were found in lower concentrations in the evaluated species compared to *Moringa oleifera* ($\bar{x}$ = 10.15 mg/QE/g^−1^) [74], *Vachellia nilotica* ($\bar{x}$ = 1.14 mg/QE/g^−1^) [75], and *Medicago sativa* ($\bar{x}$ = 6.65 mg/RE/g^−1^) [76]. Nevertheless, Cavazos et al. (2021) [77] reported antioxidant activity in *Vachellia rigidula* extracts, which was attributed to the presence of terpenes and tannins. It is important to note that limited information is available regarding the secondary metabolite profile and antioxidant potential of *Havardia pallens*. While *M. sativa* has been extensively studied, and its leaves, flowers, and seeds have been found to contain appreciable levels of phenolic compounds, flavonoids, and antioxidant activity [77,78,79].

### 4.3. Antioxidative Capacity

In diets prepared with the native species, *Havardia pallens* and *Vachellia rigidula*, the difference in antioxidant capacity values is notable compared to diets prepared with *Medicago sativa* straw, and this distinction is even more noticeable when the three plant species are not added to the diet. Therefore, the effect of bromatological characteristics and the presence of secondary compounds is observed in diets prepared with these plant species. Thus, *Havardia pallens* and *Vachellia rigidula*, when added to diets based on conventional feeds, demonstrate exceptional potential as functional and nutraceutical alternatives.

Delgadillo-Puga et al. (2019) [67] formulated diets for dairy goats with incremental inclusion levels (10%, 20%, and 30%) of *Vachellia farnesiana*, in combination with *Medicago sativa* (48–60%). These diets presented crude protein (CP) levels of 15.38%, 14.98%, and 13.85%, respectively. The impact of these formulations was assessed through milk quality, revealing that the diet containing 30% *Vachellia farnesiana* resulted in an average phenolic content of 305.5 mg/GAE/L^−1^ of milk, which was substantially higher than that obtained with the conventional diet ($\bar{x}$ = 159.4 mg/GAE/L^−1^). A similar trend was observed in antioxidant activity, with the same formulation reaching 61.8 µM/FeSO_4_/100 mL^−1^, while the conventional diet did not exhibit detectable antioxidant capacity. Based on these findings, this study proposes the strategic use of native species during important periods of the production cycle to reduce or eliminate dependence on cereals such as soybean meal or forage legumes such as *Medicago sativa*, which are often expensive or unavailable in certain regions due to market demand [5,79].

### 4.4. Phytochemical Profile

Plant species of the Fabaceae family are recognized for their broad spectrum of biological activity, which has led to the therapeutic use of more than 500 species worldwide [80]. In this context, species within the Vachellia genus exhibit potent antioxidant, antibacterial, antifungal, antitumor, antiparasitic, cytotoxic, immunomodulatory, hepatoprotective, gastroprotective, and insecticidal properties [81,82]. Specifically, methanolic extracts of *Vachellia rigidula* have demonstrated fungicidal activity against *Candida parapsilosis* [83]. These extracts have also been used as stabilizing agents for silver nanoparticles, targeting drug-resistant pathogenic bacteria in vivo [84]. Similarly, Cavazos et al. (2021) [77] reported that the presence of phenols, flavonoids, saponins, terpenes, and tannins in *Vachellia rigidula* extracts confers significant antibacterial activity against *Providencia alcalifaciens* (*p* < 0.001), *Enterococcus faecalis* (*p* < 0.01), *Staphylococcus aureus* (*p* < 0.001), and *Yersinia enterocolitica* (*p* < 0.001).

The Vachellia species analyzed in this study are characterized by high levels of flavonoids (particularly catechin and quercetin), terpenoids, and phenolic acids [85]. The inclusion of *Vachellia rigidula* in feed formulations offers a promising nutraceutical strategy, particularly for young goats, due to their antioxidant compounds, capable of scavenging and neutralizing reactive oxygen species (ROS) before they induce DNA damage, lipid peroxidation, or the onset of oxidative stress-related pathologies. Interestingly, *Vachellia rigidula* extracts are also used as active ingredients in human dietary supplements marketed for weight management. This effect is attributed to its phenethylamine content, which has been associated with stimulant, appetite-suppressing, thermogenic, and weight-reducing effects [86,87]. Therefore, the inclusion of *Vachellia rigidula* in the diet of young goats should be carefully evaluated to ensure that it does not negatively affect weight gain or other zootechnical parameters. Therefore, the selection of appropriate extraction methods and solvents is essential not only to maximize the yield of bioactive compounds but also to accurately assess their biological activity [88]. Similarly, *Havardia* spp. belongs to the Fabaceae family, and certain species of this genus (e.g., *H. albicans*) have been explored as anthelmintic alternatives due to their content of secondary metabolites such as condensed tannins, flavonoids, and saponins [89]. These compounds exert antiparasitic effects by altering energy metabolism, impairing metamorphosis, inducing non-locomotor dysfunctions, and causing epidermal damage in gastrointestinal parasites [90]. In this context, Torres-Fajardo et al. (2021) [53] reported a potent anthelmintic activity of *H. albicans* extracts against *Haemonchus contortus* throughout its developmental stages (egg, larva, and adult). Regarding the nutritional aspect, Castañeda-Ramírez et al. (2018) [91] reported that Havardia species can provide adequate values for ruminant feeding, including a crude protein content greater than 10%, metabolizable energy greater than 2.9 MJ/kg^−1^ DM, and a wide range of condensed tannin concentrations (1.0–37.6%). However, Galicia-Aguilar et al. (2012) [92] observed lower dry matter digestibility in lambs fed diets containing these shrubs. Similarly, Sandoval-Pelcastre et al. (2020) [93] reported that the tannins present in these legumes negatively affect cellulolytic bacteria, thus limiting the anaerobic fermentation of carbohydrates into volatile fatty acids and reducing the production of CO_2_ and H_2_, essential for methanogenesis [94]. *Havardia pallens* is currently being investigated as a potential forage resource, despite its historically low nutritional profile and limited use in ruminant diets [95]. The present study contributes valuable data on the chemical composition of this species, supporting future research aimed at accurately quantifying its secondary compounds, mineral content, and functional compounds, to assess its long-term sustainability for animal or even human consumption. Future research should consider the phytochemical dynamics of *Havardia pallens* throughout its phenological stages, as well as the influence of biotic and abiotic factors, to gain a comprehensive understanding of its nutritional potential and functional value. The plant species characterized in the present experiment and added to a grain-based diet, present nutritional chemical characteristics to be used in animal feed, adding them to these diets showed to provide them with in vitro properties that could affect the animals that consume them such as spermatogenesis, lactation, meat quality, immune system, blood cell production [72,96,97,98,99] so the next step to take and which is the limitation of our work, is to test them (in vivo) in growing male goats. However, within the methodology for determining whether a food has nutraceutical characteristics, the first step is an agronomic and in vitro study before it is used in animals [53].

## 5. Conclusions

The present study provides evidence that native leguminous shrubs such as *Vachellia rigidula* and *Havardia pallens*, traditionally available in the thorn scrub ecosystem of northeastern Mexico, possess significant nutraceutical potential when incorporated into functional diets for goats. The bromatological analysis revealed that *Vachellia rigidula* exhibited the highest values of acid detergent fiber (ADF) and lignin, which may initially suggest a lower digestibility; however, its exceptional content of phenolic compounds and antioxidant capacity, particularly in aqueous and methanolic extractions, highlights its role as a functional additive capable of mitigating oxidative stress and improving animal health.

The phenolic and flavonoid concentrations observed in *Vachellia rigidula* and *Havardia pallens*, though moderate when compared to other species such as *Moringa oleifera*, were sufficient to impart bioactivity in formulated diets. In particular, the DVR formulation (based on V. rigidula) showed significantly higher antioxidant capacity than both the DHP and DME diets, reinforcing the value of this species as a functional feed component. Moreover, the presence of secondary metabolites in these native legumes contributes not only to antioxidative functions but also to antimicrobial, anti-inflammatory, and anthelmintic properties previously documented in the literature.

From a zootechnical standpoint, the inclusion of *Vachellia rigidula* and *Havardia pallens* in goat diets represents a sustainable and regionally adaptable strategy, especially in semi-arid areas where conventional forages like *Medicago sativa* may be economically or ecologically limiting. The potential of *Vachellia rigidula* to influence weight gain must be carefully assessed due to its known content of phenethylamines, which are associated with thermogenic and appetite-suppressant effects in humans. Therefore, further research is required to evaluate its long-term impact on animal performance, metabolism, and reproductive parameters.

In conclusion, the findings of this study support the integration of native leguminous shrubs into caprine feeding systems as functional and nutraceutical resources. Future research should aim to deepen the characterization of their phytochemical profiles across different phenological stages and assess the bioavailability and physiological effects of their bioactive compounds in vivo. Such work would contribute to optimizing their inclusion in livestock nutrition and to advancing sustainable animal production systems in marginal environments.

## Figures and Tables

**Table 1 metabolites-15-00457-t001:** Diets formulated with *V. rigidula*, *H. pallens*, and *M. sativa*.

Ingredient (%)	DME	DVR	DHP
*M. sativa*	35	-	-
*V. rigidula*	-	35	-
*H. Pallens*	-	-	35
Ground sorghum	35	35	35
Sorghum straw	14	14	14
Molasses	7	7	7
Soybean meal	8	8	8
Minerals	1	1	1
Salt	0.10	0.10	0.10
Total	100	100	100

DME: diet with *M. sativa*; DVR: diet with *V. rigidula*; DHP: diet with *H. pallens.*

**Table 2 metabolites-15-00457-t002:** Nutrient content (%) of two native species (*V. rigidula, H. pallens*) and a forage legume (*M. sativa*) and of diets prepared with these species.

Plant	ADF	NDF	CP	ASH	Lignin
*H. pallens*	31.43 ± 4.75 ^ab^	48.3 ± 3.05 ^a^	20.67 ± 1.69 ^ab^	7.39 ± 0.37 ^c^	9.91 ± 1.03 ^b^
*V. rigidula*	40.48 ± 4.71 ^a^	49.72 ± 7.90 ^a^	12.84 ± 2.21 ^c^	5.92 ± 0.70 ^c^	21.4 ± 3.43 ^a^
*M. sativa*	24.44 ± 1.81 ^bc^	46.73 ± 6.21 ^ab^	22.66 ± 2.13 ^a^	8.3 ± 2.08 ^bc^	8.70 ± 0.61 ^b^
DVR	21.18 ± 0.54 ^bc^	37.47 ± 0.90 ^bc^	13.4 ± 0.05 ^c^	10.15 ± 0.02 ^ab^	*
DHP	14.86 ± 0.25 ^c^	31.54 ± 1.28 ^c^	14.25 ± 0.35 ^c^	11.33 ± 0.10 ^a^	*
DME	15.81 ± 0.36 ^c^	31.38 ± 0.17 ^c^	16.7 ± 0.01 ^bc^	11.86 ± 0.09 ^a^	*

ADF: acid detergent fiber; NDF: neutral detergent fiber; CP: crude protein; DVR: diet with *V. rigidula*; DHP: diet with H. *pallens*; DME: diet with *M. sativa*. * Lignin content not measured in diets. Means with the same letter do not differ statistically (Tukey *p* < 0.05).

**Table 3 metabolites-15-00457-t003:** Total phenols and flavonoids in *V. rigidula*, *H. pallens*, and *M. sativa*.

Type of Extraction	Phenols (mg/GAE/g^−1^)	Flavonoids (mg/QE/g^−1^)
*V. rigidula* aqueous extraction	18.22 ± 0.11 ^a^	1.16 ± 0.0049
*H. pallens* aqueous extraction	11.23 ± 0.02 ^abc^	0.88 ± 0.0048
*M. sativa* aqueous extraction	8.68 ± 0.03 ^bc^	9.422 ± 0.4833
*V. rigidula* methanolic extraction	16.82 ± 0.29 ^ab^	0.295 ± 0.0119
*H. pallens* methanolic extraction	8.16 ± 0.09 ^bc^	0.302 ± 0.0047
*M. sativa* methanolic extraction	7.61 ± 0.10 ^c^	1.285 ± 0.0124
*V. rigidula* ethanolic extraction	17.04 ± 0.08 ^ab^	0.311 ± 0.0049
*H. pallens* ethanolic extraction	9.59 ± 0.05 ^abc^	0.336 ± 0.0019
*M. sativa* ethanolic extraction	7.61 ± 0.10 ^c^	1.260 ± 0.01

GAE: gallic acid equivalent; QE: quercetin equivalent. Means with the same letter within each column do not differ statistically (Tukey *p* < 0.05).

**Table 4 metabolites-15-00457-t004:** Concentration of total phenols (g/GAE/g^−1^), total flavonoids (g/QE/kg^−1^) in diets formulated with two native species (*V. rigidula* and *H. pallens*) and a forage legume (*M. sativa*).

Secondary Compounds	DME	DVR	DHP	SEM	*p*-Value
Phenols	22 ^b^	57 ^a^	47 ^a^	0.006	0.007
Flavonoids	210 ^ab^	230 ^a^	180 ^b^	0.008	0.005

DME: *Medicago sativa*, DVR: *Vachellia rigidula*; DHP: *Harvardia pallens*; SEM: Standard error of the mean, GAE: Gallic acid equivalent, QE: Quercetin equivalent. Means with the same letter within each column do not differ statistically (Tukey *p* < 0.05).

**Table 5 metabolites-15-00457-t005:** Antioxidative capacity.

Species	Antioxidative Capacity (μmol/L /TE/g^−1^)
*V. rigidula*	144,711.53 ± 10,442.87 ^a^
*H. pallens*	16,800.21 ± 1287.84 ^b^
*M. sativa*	11,701.92 ± 319.16 ^b^

TE: Trolox equivalent; mmol: millimoles. Means with the same letter within each column do not differ statistically (Tukey *p* < 0.05).

**Table 6 metabolites-15-00457-t006:** Antioxidant capacity of diets formulated with *V. rigidula*, *H. pallens*, and *M. sativa* and without inclusion.

Type of Extraction by Species	μmol/ET/g^−1^
DVR aqueous extraction	147,966.66 ± 10,315.38 ^a^
DHP aqueous extraction	17,949.30 ± 1441.13 ^b^
DME aqueous extraction	19,143.74 ± 2908.89 ^b^
DVR methanol extraction	139,847.44 ± 11,097.05 ^a^
DHP methanol extraction	10,591.66 ± 474.040 ^b^
DME methanol extraction	8206.24 ± 1460.67 ^b^
Mixture without *V. rigidula* aqueous extraction	9204.06 ± 978.95 ^b^
Mixture without *H. pallens* aqueous extraction	8154.91 ± 481.25 ^b^
Mixture without *M. sativa* aqueous extraction	9204.06 ± 978.95 ^b^
Mixture without *V. rigidula* methanol extraction	9159.19 ± 25.90 ^b^
Mixture without *H. pallens* methanol extraction	7772.43 ± 381.45 ^b^
Mixture without *M. sativa* methanol extraction	9159.18 ± 25.90 ^b^

DME: diet with *M. sativa*; DVR: diet with *V. rigidula*; DHP: diet with *H. pallens*; ET: Trolox equivalent; Means with the same letter within each column do not differ statistically (Tukey *p* < 0.05).

**Table 7 metabolites-15-00457-t007:** *V. rigidula* phytochemical identification through HPLC-ESI-MS.

RT (min)	[M-H]-(*m*/*z*)	Compounds	Solvent	Family	Effects	Reference
5.369	376.8	3,4-DHPEA-EA	A, E, M	Tyrosols	Antioxidant	[21]
18.417	314.9	Nepetin	A, E, M	Methoxyflavones	Antibacterial, antioxidant, anti-inflammatory, cytotoxic	[22]
22.852	576.8	Procyanidin dimer B1	A	Proanthocyanidin dimers	Antioxidant, antimicrobial, anthelmintic, anti-inflammatory, gastro-protective, and insecticidal activities.	[23]
23.893	386.9	Medioresinol	(A)	Lignans	Potential for AChE inhibition and antimicrobial	[24,25]
25.567	288.9	(+)-Catechin	A, E, M	Catechins	Antioxidant, antimicrobial, anti-acne, anthelmintic, anti-inflammatory, gastro-protective, and insecticidal activities.	[26]
27.847	562.8	Apigenin arabinoside-glucoside	A, E, M	Flavones	Antioxidant	[27]
31.107	462.8	Myricetin 3-O-rhamnoside	A, E, M	Flavonols	Antioxidant, antifungal, antimicrobial.	[28]
34.75	446.8	Luteolin 6-C-glucoside	A, E, M	Flavones	Insecticide (pests by influencing their behavior, growth, and development)	[29]
7.827	340.8	Caffeic acid 4-O-glucoside	E	Hydroxycinnamic acids	Antioxidant (anti-melanogenesis and anti-tyrosinase), insecticide	[30]
18.747	592.7	Apigenin 6,8-di-C-glucoside	E	Flavones	Antioxidant, anti-cancer and insecticide	[31,32]
21.758	288.8	(−)-Epicatechin	E	Catechins	Antioxidant, anti-inflammatory, anti-cancer, antidiabetic, cardio and neuroprotective.	[33]
27.614	562.8	Apigenin galactoside-arabinoside	M	Flavones	Antioxidant	[34]

RT: Retention time. [M-H] (*m*/*z*): mass. Solvent: (A) aqueous; (E) ethanol; (M) methanol.

**Table 8 metabolites-15-00457-t008:** *H. pallens* phytochemical identification through HPLC-ESI-MS.

RT (min)	[M-H]-(*m*/*z*)	Compounds	Solvent	Family	Effects	Reference
5.079	316.8	Myricetin	A	Flavonols	Antioxidant, antibacterial, antiviral, anti-inflammatory, anti-allergic.	[34,35]
17.905	354.8	Ferulic acid 4-O-glucoside	A, E, M	Methoxycinnamic acids	Antioxidant, antimicrobial, anti-inflammatory, anticancer, cardio, gastro and neuroprotective	[36]
26.993	370.8	Sinensetin	A	Methoxyflavones	Antioxidant, anti-inflammatory, antimicrobial, anti-obesity, anti- dementia and vasorelaxant activities.	[37]
34.851	518.9	5,3′,4′-Trihydroxy-3-methoxy-6:7-methylenedioxyflavone 4′-O-glucuronide	A, E, M	Methoxyflavonols	Antioxidant	[26]
6.405	341	Caffeic acid 4-O-glucoside	A, E, M	Hydroxycinnamic acids	Antioxidant (anti-melanogenesis and anti-tyrosinase), insecticide	[30,38]
38.99	595	Eriocitrin	A, E, M	Flavanones	Antioxidant, antitumor, anti-allergic, antidiabetic and anti-inflammatory	[39]
40.858	595	Neoriocitrin	A, E	Flavanones	Antioxidant	[40]
24.596	354.9	Feruloyl glucose	M	Methoxycinnamic acids	Antioxidant	[41]
39.613	463	Myricetin 3-O-rhamnoside	M	Flavanones	Antioxidant, antifungal, antimicrobial.	[42]
41.385	505	Peonidin 3-O-(6″-acetyl-glucoside)	M	Flavonols	Antioxidant, neuroprotective, anti-inflammatory.	[43]

RT: Retention time. [M-H] (*m*/*z*); mass. Solvent; (A); aqueos. (E); ethanol. (M); methanol.

## Data Availability

All the authors confirm that the data supporting the findings of this study are presented in the tables in the manuscript.

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
