# Peer review of "Nutraceutical Potential of Havardia pallens and Vachellia rigidula in the Diet Formulation for Male Goat"

_metabolites, 2025, doi:10.3390/metabo15070457_

Round 1
Reviewer 1 Report
Comments and Suggestions for Authors
The manuscript investigates the nutraceutical potential of two native legumes, Havardia pallens and Acacia rigidula, in goat diets. The study is well-structured, with clear objectives and methodologies. However, several issues require attention to improve clarity, accuracy, and adherence to academic standards.
The manuscript occasionally shifts between past and present tense (e.g., "The samples were analyzed" vs. "This study proposes"). Ensure consistent use of past tense for methods/results and present tense for established facts.
Replace informal phrases like "must be noted" with "it should be noted."
Use "however" instead of "in contrast" when contrasting within the same sentence (e.g., "In contrast, A. rigidula exhibited..." → "However, A. rigidula exhibited...").
Avoid repetitive phrases (e.g., "in the present study" appears frequently; replace with synonyms like "this research").
Correct usage of prepositions (e.g., "determined by two colorimetric methods" → "determined using two colorimetric methods").
Table 1: The method for nutrients measuring should be explain in the Method section. Clarify the footnote "*" (e.g., "Lignin content not measured in diets"). Use superscript letters (a, b, c) for statistical notations instead of mixed formatting (e.g., "± 4.75abab").
Table 3: Standardize decimal places (e.g., "17.04 ± 0.08" vs. "7.61 ± 0.10").
Table 5: The antioxidant capacity value for A. rigidula (144,711.53 mmol TE/g) is unusually high. Verify the units (e.g., µmol vs. mmol) and ensure consistency with FRAP method descriptions.
Specify the statistical significance of results (e.g., "P<0.05") in the abstract.
Strengthen the rationale by emphasizing the novelty of using H. pallens and A. rigidula compared to existing studies on other legumes.
Section 2.3: Clarify "partial dry matter (DM)" vs. "total moisture."
Section 2.9: Define "sorghum soca" for non-specialist readers.
Avoid repeating data between tables and text. Use the text to highlight trends, not duplicate table content.
Compare findings with recent studies (e.g., post-2020 references are sparse).
Address the contradiction between A. rigidula's high lignin content and its proposed digestibility.
Statistical Analysis: Specify the ANOVA model used (e.g., one-way, factorial).
References: Ensure journal-specific formatting (e.g., italicize journal names, standardize DOI links). Update references to include recent studies (e.g., only 4/86 references are post-2020).
Recommendations
- Revise language for clarity and consistency.
- Validate antioxidant capacity calculations and units.
- Expand the discussion to address limitations (e.g., lack of in vivo validation).
- Update references and ensure all citations are complete.
- Simplify tables by removing redundant data and standardizing formatting.
The study offers valuable insights into the nutraceutical potential of underutilized legumes. Addressing the above issues will enhance its rigor and readiness for publication.
Comments on the Quality of English LanguageRevise language for clarity and consistency
Author Response
- Quality of English Language (x) The English could be improved to more clearly express the research. Response: The manuscript's writing was improved, the English expression was modified, and the statements that were in the present tense were also changed to past tense. This was done throughout the manuscript.
- Replace informal phrases like "must be noted" with "it should be noted." Response: The replacement was made on line 402
- Use "however" instead of "in contrast" when contrasting within the same sentence (e.g., "In contrast, A. rigidula exhibited..." → "However, A. rigidula exhibited..."). Response: "In contrast" was replaced by "however" on lines 231, 245, 306, 315
- Avoid repetitive phrases (e.g., "in the present study" appears frequently; replace with synonyms like "this research"). Response: The wording was changed so that the phrase "in the present study" is no longer found in the manuscript.
- Correct usage of prepositions (e.g., "determined by two colorimetric methods" → "determined using two colorimetric methods"). Response: The wording of the materials and methods section has been modified so that the phrase "determined by two colorimetric methods" is no longer included.
- Table 1: The method for nutrients measuring should be explain in the Method section. Clarify the footnote "*" (e.g., "Lignin content not measured in diets"). Use superscript letters (a, b, c) for statistical notations instead of mixed formatting (e.g., "± 4.75abab"). Response: The order of appearance of the tables was modified to be more consistent with the narrative of the manuscript, so Table 1 is now Table 2. The wording of the methodology for determining the bromatological elements was modified, and the citation AOAC (2000) was added. Therefore, lines 104-113 were modified. In Table 2, all the literals are in superscript.
- Table 3: Standardize decimal places (e.g., "17.04 ± 0.08" vs. "7.61 ± 0.10"). Response: The modification of table 3 was made for the position of the decimals
- Table 5: The antioxidant capacity value for A. rigidula (144,711.53 mmol TE/g) is unusually high. Verify the units (e.g., µmol vs. mmol) and ensure consistency with FRAP method descriptions. Response: The error was addressed, and millimoles were changed to micromoles and so on throughout the manuscript.
- Specify the statistical significance of results (e.g., "P<0.05") in the abstract. Response: Changes were made to the abstract.
-
Strengthen the rationale by emphasizing the novelty of using H. pallens and A. rigidula compared to existing studies on other legumes.
-
Section 2.3: Clarify "partial dry matter (DM)" vs. "total moisture." The paragraph was modified on lines 105 to 114, as well as on lines 192 to 196
-
Section 2.9: Define "sorghum soca" for non-specialist readers. Response: The wording has been revised to include a brief definition of “sorghum soca” to facilitate understanding among non-specialist readers. It is now specified that this term refers to the regrowth of the sorghum crop after the first harvest, which is commonly used as forage in animal feeding systems.
-
Avoid repeating data between tables and text. Use the text to highlight trends, not duplicate table content.
-
Compare findings with recent studies (e.g., post-2020 references are sparse). Changes were made to lines 312 to 314, 318 to 326, 332 to 344, 453 to 460 and current quotes were added to the discussion section.
-
Address the contradiction between A. rigidula's high lignin content and its proposed digestibility. The fragment about digestibility was removed
-
Statistical Analysis: Specify the ANOVA model used (e.g., one-way, factorial). Response: The manuscript has been revised to specify that a one-way analysis of variance (ANOVA) was performed for each of the evaluated variables. This clarification was included to enhance methodological precision and facilitate the interpretation of the statistical analysis applied.
-
References: Ensure journal-specific formatting (e.g., italicize journal names, standardize DOI links). Update references to include recent studies (e.g., only 4/86 references are post-2020). The references were updated.

Reviewer 2 Report
Comments and Suggestions for Authors
The article is devoted to the nutritional content of goat diets, including plants of two representatives of the Fabaceae family, Havardia pallens and Acacia rigidula, in areas of thorn scrub ecosystem of Mexico. The article is described topics of zootechnics, the diversity of goat feed and their composition. Publication will also be of interest to readers interested in phenolic compounds of Fabaceae. The research methods correspond to the studied problems, the results and conclusions are adequate to the methods used. Plagiarism and excessive self-citation were not detected (self-citation - 5 of 83 cited references). The cited references correspond to the topics discussed in the article.
Note: the list of references is in sentence case, but references 3, 43, 51 in title case. It is necessary to unify.
The manuscript is suitable to publication in its current form.
Author Response
The observations in the bibliography were resolved.
